# Ultra-High-Sensitivity Humidity Fiber Sensor Based on Harmonic Vernier Effect in Cascaded FPI

**DOI:** 10.3390/s22134816

**Published:** 2022-06-25

**Authors:** Cheng Zhou, Yanjun Song, Qian Zhou, Jiajun Tian, Yong Yao

**Affiliations:** School of Electronic and Information Engineering, Harbin Institute of Technology, Shenzhen 518055, China; 20b352001@stu.hit.edu.cn (C.Z.); 20s152137@stu.hit.edu.cn (Y.S.); 19s052006@stu.hit.edu.cn (Q.Z.); yaoyong@hit.edu.cn (Y.Y.)

**Keywords:** Fabry–Perot interferometer, chitosan film sensor, optical fiber humidity sensor, harmonic Vernier effect

## Abstract

In this study, an ultra-high-sensitivity fiber humidity sensor with a chitosan film cascaded Fabry–Perot interferometer (FPI) based on the harmonic Vernier effect (HVE) is proposed and demonstrated. The proposed sensor can break the limitation of the strict optical path length matching condition in a traditional Vernier effect (TVE) FPI to achieve ultra-high sensitivity through the adjustment of the harmonic order of the HVE FPI. The intersection of the internal envelope tracking method allows spectra demodulation to no longer be limited by the size of the FSR of the FPI. The sensitivity of the proposed sensor is −83.77 nm/%RH, with a magnification of −53.98 times. This work acts as an excellent guide in the fiber sensing field for the further achievement of ultra-high sensitivity.

## 1. Introduction

The humidity in daily life refers to relative humidity (RH), which means the percentage of the amount of water vapor contained in the air and the amount of saturated water vapor that can exist in the air under the same conditions [1]. Humidity detection plays an important role in agricultural production, meteorology, chemical processing, food storage, medical identification, and structural health monitoring [2,3]. In recent years, many optical fiber sensors have been studied and reported due to their anti-electromagnetic interference, fast response, and high sensitivity [4,5,6,7,8,9,10,11,12,13,14,15,16,17,18,19,20]. These sensors are based on a variety of structures, such as the fiber Fabry–Perot interferometer (FPI) [4], fiber tip [5], fiber Bragg grating [7,8], Mach–Zehnder tapered-fiber interferometer [9], and Michelson interferometer [10].

Among them, the FPI has important application prospects due to its compact structure and easy fabrication [11,12,13,14,15,16,17,18,19,20]. To further improve RH sensitivity, the FP cavity is usually made of hygroscopic material. Yang et al. used Al_2_O_3_ film to form a FP cavity for sensitivity enhancement [12]. Their work found FP constructed by all hygroscopic material could obtain higher sensitivity for efficient use of coefficient of expansion of hygroscopic material [13,14]. A study by Zhang et al. presented the spider dragline silk-based RH sensor and the RH sensitivity could be average 17.2 nm/%RH form range of 58–98 %RH [15]. Research shows that the shorter the FP cavity length, the higher the sensitivity of the sensor [16]. Therefore, sensors are usually prepared as a thin-film FP with a typical thickness below 50 μm [4,14]. Another method to improve sensitivity is using TVE by connecting two FP cavities together. If the optical path length (*OPL*) matching condition is met, i.e., the *OPL*s of the two FPs are sufficiently close but not equal, their spectra are superimposed to generate a new spectrum with an envelope. The period of this envelope is the least common multiple of the two FPs. The sensitivity can be amplified times the least common multiple by spectral tracking of the envelope. These two FPs in TVE FPI can be arranged by both cascaded and parallel structure [21,22]. However, a thin-film sensing FP is usually accompanied by an additional air FP cavity in fabrications [21,23], which will break the *OPL* matching condition of TVE, and lead in information aliasing if parallel structure is used. The use of the cascaded structure and the rational use of the air cavity as the reference cavity in the thin-film FPI can realize a TVE FPI with sensitivity amplification. However, high sensitivity magnification is still usually limited in the TVE FPI.

Firstly, the TVE has a very harsh *OPL* matching condition. According to this matching principle, the magnification factor (*M*-factor) of the sensitivity can be defined as the free spectral range (FSR) ratio between the envelope of the TVE spectrum and the spectrum of the sensing FP. This is defined as follows [23]:(1)$$M_{0}=\frac{n_{2}L_{2}}{n_{1}L_{1}-n_{2}L_{2}}=\frac{n_{2}L_{2}}{\Delta },$$
where *L_x_* and *n_x_* (*x* = 1, 2) are the length and refractive index of reference cavity FP1 and sensing cavity FP2, respectively, and Δ is defined as the *OPL* mismatch between the two FPs. It can be seen in Equation (1) that the smaller the Δ, the bigger the *M*. However, in a practical preparation process, such as fiber cleaving, splicing, and etching, machining error is inevitably induced, resulting in the Δ deviating from the designed value and a reduction of the *M*-factor.

As an example, we consider a cascaded cavity TVE FPI. If the *OPL* of the sensing cavity is 50 μm and the preset *M*-factor is 48.08 times, the Δ should be 1.04 μm, according to Equation (1). However, if the machining error of 2 μm is induced, the *M*-factor will be reduced from 48.08 times to 16.45 times, which is only 34.2% of the ideal state, shown in Figure 1. The high *M*-factor usually needs a Δ under a sub-micron level, however, in the actual process, the micron-level error is hard to avoid, so most of the *M*-factor of the TVE FPI is less than 30 times [22,23,24,25,26]. Second, an ultra-high-sensitivity TVE FPI is also limited in the demodulation method using envelope tracking. The TVE FPI needs to identify the extreme point of the envelope as a tracking point, so there must be at least one full-period envelope in the visible range of demodulation instrument. However, the thin-film FP has a large FSR (e.g., an agar FP with an FSR of 127 nm [4]). Therefore, even if ideal *OPL* matching is achieved, the FSR of the envelope will be further enlarged and be far beyond the wavelength range of the broadband light source (e.g., the broadband light source: Fiber Lake, ASE light source, wavelength range: around 450 nm), resulting in a failed measurement.

Recently, a new theory of harmonic Vernier effect (HVE) has been reported [27,28], which could further magnify the envelope sensitivity by *i* times compared with TVE. It could be achieved by increasing the *OPL* of first interferometers by a multiple (*i*-times) of *OPL* of the second interferometer. Since the magnification is multiplied by the basic magnification and the *i* order, the ideal magnification could be more easily obtained with higher fabrication tolerance. A gas pressure sensor reported by Wu recently verified the theory of HEV [29]. Such a practical theory has not been applied to a cascaded FP structure to enhance the RH sensitivity.

In this work, an ultra-high-sensitivity thin-film (chitosan) RH fiber sensor based on the HVE in a cascaded FPI is proposed and demonstrated. Different from that of the TVE FPI, the *M*-factor of the proposed HVE FPI not only depends on Δ, but also the harmonic order. Therefore, a high *M*-factor can be easily achieved by increasing the harmonic order. This characteristic also results in greater *OPL* matching tolerance, resulting in easier and more convenient realization of an ultra-high-sensitivity HVE FPI compared to that of a TVE FPI. For the demodulation, we use the internal envelope fitting method to track the intersection of the internal envelope, which solves the problem of the invisible envelope extreme point caused by the large FSR. Experimental results show that the proposed sensor can achieve an *M*-factor of more than 50 times for RH sensing. The method provides a good reference for high-sensitivity sensing in the fiber sensing field.

## 2. Experimental Setup and Working Principle

A schematic diagram of the proposed HVE FPI sensor is shown in Figure 2. The HVE FPI sensor includes a single-mode fiber (SMF), a chitosan film, and a fiber tube. Because of the different refractive indexes, there are three reflective surfaces, labeled M1, M2, and M3. The chitosan cavity, FP2 (M2–M3), is the sensing cavity (*n*_2_ = 1.52), and the air cavity, FP1 (M1–M2), is the reference cavity (*n*_1_ = 1).

Since the reflectivity at each reflecting surface is less than 4%, the reflection beam can be simplified as three-beam interference. The total reflected intensity can be expressed as follows [21]:(2)$$I_{r}=E_{in}^{2}[A+B\cos\left[2(\phi_{1}+\phi_{2})\right]+C\cos\left(2\phi_{2}\right)+D],$$
where
(3)$$\begin{array}{l} A=R_{1}+{\left(1-\alpha_{1}\right)}^{2}{\left(1-R_{1}\right)}^{2}R_{2}+{\left(1-\alpha_{1}\right)}^{2}{\left(1-\alpha_{2}\right)}^{2}{\left(1-R_{1}\right)}^{2}R_{3},\\ B=2\sqrt{R_{1}R_{3}}(1-\alpha_{1})(1-\alpha_{2})(1-R_{1})(1-R_{2}),\\ C=2\sqrt{R_{2}R_{3}}{\left(1-\alpha_{1}\right)}^{2}(1-\alpha_{2}){\left(1-R_{1}\right)}^{2}(1-R_{2}),\\ D=2\sqrt{R_{1}R_{2}}(1-\alpha_{1})(1-R_{1}) \end{array}$$

In Equation (3), $R_{m}\left(m=1,2,3\right)$ is the reflectivity of three surfaces; $\alpha_{n}\left(n=1,2\right)$ is the loss of the reflector; $\phi_{1}=2\pi n_{1}L_{1}/\lambda$ and $\phi_{2}=2\pi n_{2}L_{2}/\lambda$ are the phase delay of FP1 and FP2, respectively; and λ is the wavelength in vacuum.

To study the formation mechanism of HVE FPI and discuss the *OPL* matching condition of FP1 and FP2, it is assumed that *OPL*_1_ > *OPL*_2_ with the following relationship [28]:(4)$$OPL_{1}=n_{1}L_{1}=(i+1)n_{2}L_{2}+\Delta ,$$
where *i* is an integer defined as the harmonic order of HVE. The harmonic order *i* depends on the ratio of the two FP *OPL*s and is expressed as follows:(5)$$i=\left\lfloor \frac{n_{1}L_{1}}{n_{2}L_{2}}\right\rfloor -1,$$
where the mathematical symbol ⌊X⌋ indicates the rounding down of *X*. When i≠0, according to the *OPL* relationship, although the TVE condition is not met, the (*i* + 1) times *FSR_1_* is similar to the *FSR_2_*, so there is still a least common multiple between two FPs. The total spectrum function of the sensor can be rearranged as follows [18]:(6)$$I_{r}=E_{in}^{2}[A+\sqrt{B^{2}+D^{2}+2BD\cos(2\phi_{2})}sin(2\phi_{1}+\phi_{2}+\theta )+C\cos(2\phi_{2})],$$
where $tg\theta =-[B+D\cos(2\phi_{2}+2\phi_{1})]/[D\cos(2\phi_{1}-2\phi_{2}]$ and the $\cos(\phi_{2})$ term corresponds to the low-frequency envelope. The term of $sin(2\phi_{1}+\phi_{2}+\theta )$ corresponds to the high-frequency fringes (HFFs), which are modulated by $\cos(\phi_{2})$. Moreover, $\phi_{2}$ also determines the phase information of the upper envelope, which will be discussed in detail in Figure 3. Therefore, the upper envelope function of the reflection spectrum superposed by all terms can be given as follows:(7)$$I_{r}=m\cos(2\phi_{2}),$$
where *m* is the amplitude.

According to Equations (4) and (5), both *i* and Δ can be adjusted by tuning the *OPL* ratio between FP1 and FP2. To investigate the difference between TVE and HVE FPIs, a different *OPL* ratio is chosen and studied. The calculated spectra for individual FP1 and FP2 at *i* = 0 (*L*_1_ = 34.2 μm, *L*_2_ = 20 μm, and Δ = 3.8 μm) are shown in Figure 3a. The wavelengths of the different peaks are labeled as $\lambda_{p}^{k}$, where *p* = 1, 2 is the number of the FP and *k* is the number of the peak. The FSRs of two FPs are close (FSR_1_ = 24 nm, FSR_2_ = 27 nm), and the magnification k of the least common multiple of the two FPs is the peak number between two overlapped peaks of the two FPs. The wavelength relationship of overlapped peaks can be expressed as follows:(8)$$\lambda_{1}^{k+1}=\lambda_{2}^{k},$$
where $k=\left|\frac{FSR_{2}}{FSR_{1}-FSR_{2}}\right|$ (in the calculation, *k =* 8). Figure 3a also shows the entire calculated spectrum of the cascaded FPI, wherein there is a large envelope (labeled as env0) from fitting all HFF peaks in turn. This is the typical TVE spectrum. The expression for the *FSR* of env0 is [27,28]:(9)$$FSR_{env0}=kFSR_{1}=\frac{FSR_{1}FSR_{2}}{FSR_{2}-FSR_{1}}=\frac{\lambda_{1}^{0}\lambda_{1}^{k+1}}{2\Delta },$$

The TVE FPI will evolve into HVE FPI if *i* > 0. We increased the length of *L*_1_ to 64.6 μm, thus *i* = 1 and Δ remain as 3.8 μm. Figure 3b shows the spectra of FP1, FP2, and cascaded HVE FPI. Now, *FSR*_1_ becomes 10.3 nm and the wavelength of the overlapped peaks of FP1 and FP2 can be expressed as follows:(10)$$\lambda_{1}^{(i+1)k+1}=\lambda_{2}^{k},$$
where $k=\left|\frac{FSR_{2}}{FSR_{1}-iFSR_{2}{}^{i}}\right|$. If we fit all the HFF peaks of cascaded FPI spectrum in turn, an upper envelope will be obtained, which differs from env0 of the TVE shown in Figure 3a. The FSR of the upper envelope is 27 nm, which is exactly consistent with that of FP2 and Equation (7). If we further fit the peaks of the upper envelope, an outer envelope with the same FSR as env0 of TVE will be obtained. The expression for the FSR of the outer envelope is [27,28]:(11)$$FSR_{out}=\frac{FSR_{1}^{i}FSR_{2}}{FSR_{2}-(i+1)FSR_{1}},$$

Another remarkable difference from TVE spectrum is that if we fit the HFF peaks with two peaks as intervals, there are two obvious internal envelopes with the same FSR, which is twice as much as that of the outer envelope. A 1-order HVE FPI spectrum is obtained, as shown in Figure 3b. We can find that there are three HFFs in an upper envelope period in this spectrum. At the same time, there are two obvious internal envelope curves obtained by taking points at intervals. Moreover, the FSR of the inner envelope is twice that of the outer envelope.

For a more comprehensive understanding of the formation of the inner envelope, we continued to increase the length of *L_1_*. For example, Figure 4a shows 4-order HVE FPI spectrum, in which *L_1_* = 155.8 μm (*i* = 4, Δ = 3.8μm), and there are six HFFs in an upper envelope period. We take these fringe peaks as the starting point and take points at an interval of five points for envelope fitting. In Figure 4b, five inner envelope curves appear (the inner envelope curve starting from the first HFF coincides with the line starting from the sixth). It is not difficult to find that for the *i*-order HVE FPI, (*i* + 1) internal envelopes can be obtained by fitting the HFF peaks with (*i* + 1) peaks as intervals. The FSR of the internal envelope is (*i* + 1) times that of the outer envelope, which can be expressed as [28]:(12)$$FSR_{\mathrm{int}}=\frac{(i+1)FSR_{1}^{i}FSR_{2}}{(i+1)FSR_{1}^{i}-FSR_{2}}=(i+1)FSR_{out},$$

The simulation spectrum shows that when the internal envelope FSR is too large to track, there are still many intersection points of the internal envelope. These intersection points can be selected as the visible tracking points so that the demodulation is no longer limited by the large FSR, thus solving the problem in TVE FPI demodulation.

The *M*-factor of *i*-order HVE FPI can be redefined as the FSR ratio of the internal envelope to the spectrum of the sensitive cavity FP2, expressed as:(13)$$M^{i}=\frac{FSR_{\mathrm{int}}}{FSR_{2}}=(i+1)\frac{n_{2}L_{2}}{\Delta }=(i+1)M,$$

Therefore, according to Equations (1) and (13), even with the same Δ, the *M*-factor of HVE FPI can be (*i* + 1) times the TVE FPI, while the tolerance of machining error for *i* is far greater than that of Δ. This is because the interval of *i* in the *OPL* of FP2 is usually tens of microns, while the Δ is usually micron- or submicron-level, as mentioned before. It should be noted that in the cascaded FPI, the sensitivity of the upper envelope reflects the sensitivity of FP2 [12], so the *M*-factor of HVE FPI in experiment can be obtained through dividing sensitivity of the internal envelope or intersections by the sensitivity of the upper envelope.

Figure 5 shows the spectral change of upper and internal envelopes when n_2_ changes from 1.520 to 1.522. It can be seen that the upper envelope drifts 2 nm, while the internal envelope intersection (env1 and env2) drifts −80 nm, making an *M*-factor of −40 times, which is consistent with Equation (13).

The fabrication method of the proposed sensor is shown in Figure 6. We prepared a 1% concentration of chitosan (C105803, Aladdin Reagent (Shanghai) Co., Ltd, Shanghai, China) acetic acid (A116166, Aladdin Reagent (Shanghai) Co., Ltd, Shanghai, China) solution and dipped an HCF (TSP150375, Polymicro Technologies, AZ, USA)) with an inside diameter of 135 μm and an outside diameter of 363 μm into the chitosan solution to make the thin chitosan film (Figure 6a). The lower end of the HCF was controlled to just touch the solution. The solution entered the HCF owing to the capillarity effect, and the height of the liquid column increased over dipping time. The HCF was then held vertically and the liquid column shrank to become a chitosan film inside the HCF after air-drying for 24 h, as shown in Figure 6b. If we set the dipping time within the limit from 0.5 to 3 s, the linear length of the chitosan film would be 1–10 μm. Therefore, the proposed method can effectively guarantee an ideal chitosan film thickness by controlling the dipping time. Two SMF sections could be obtained after cleaving the SMF, as shown in Figure 6c. Then, we inserted one end of the SMF gradually into the HCF by operating the motorized precision translation stage of the splicer, as shown in Figure 6d. The length of the air cavity could be controlled by the precision translation stage under a microscope, which is helpful to control the *OPL* ratio and investigate experimentally the characteristic of HVE FPI with different *i*-orders. Finally, we dropped glue (LOCTITE AA3321, AA3321, Korea Loctite (Shanghai) Co., Ltd, Shanghai, China) on the gap of HCF and SMF to stabilize the structure. This sensor was then placed under a violet light (wavelength: 395 nm).

Theoretically, the higher the value of *i*, the higher the harmonic orders, causing a higher inner envelope sensitivity. However, the loss of air cavity becomes larger with the increase of *OPL_1_*, which could decrease spectral contrast to affect the inner envelope fitting. The thinner the chitosan film is, the higher the basic sensitivity obtained. Meanwhile, the thinner the chitosan film is, the larger the FSR of the chitosan cavity would be. In experiment, the wavelength range is around 400 nm. According to FSR equation $FSR=\frac{\lambda^{2}}{2nL}$, chitosan film of 10 um corresponds to approximately 85 nm, which means that only four more FSRs showed up in the OSA. One FSR corresponds to one inner envelope fitting point, and few fitting points affect the envelope reduction accuracy. Therefore, these two would commonly limit the harmonic orders. In order to ensure high basic sensitivity and a suitable number of envelope fitting points, we chose the thickness of chitosan film as 20 μm and the length of the air cavity as 155 μm for optimization length according to Equation (4).

The fabricated sensor and its reflection spectrum are shown in Figure 7a, b, respectively. The *L_1_* and *L_2_* are 155 μm and 19.8 μm, respectively. It can be seen that the experimental spectrum is similar to the results in Figure 4a.

The RH sensing experimental setup is shown in Figure 8, including a broadband light source (ASE light source, Shenzhen Fiberlake Technology Co., Led, Shenzhen, China, wavelength range: 1200–1700 nm), an optical fiber circulator, an OSA (AQ6370C, Yokogawa, ANDO, USA, wavelength range: 600–1700 nm, highest resolution: 0.02 nm), a sensor probe, and an adjustable temperature and RH test chamber (CK-22G, Dongguan Kingjo Environmental Testing Equipment Co., Ltd, China, RH range: 20–98 %RH, humidity deviation: ±1% RH, Temperature range: −20~150 °C, size of inner box in chamber: 500 × 500 × 600 mm^3^). In the experiment, we set the demodulation wavelength range of OSA as 1250–1630 nm.

## 3. Experimental Results and Discussion

### 3.1. Sensitivity

In order to test the RH sensitivity of the proposed sensor, we put the sensor into the above device. The temperature of the test chamber was maintained as 25 °C while the RH level was set from 46 %RH to 60 %RH, with 2 %RH as the interval. Figure 9a shows the upper envelope spectral drift diagram of the sensor under five RH values. It can be found that as the RH increases, the upper envelope in the spectrum has a significant red shift. In monitoring the increase and decrease of RH in the same environment, the results show that the sensor has a good linear response, as shown in Figure 9b.

In the observed range, the upper envelope response sensitivity is 1.58217 nm/%RH. After linear fitting, the change of the peak position under the same RH does not exceed 0.15 nm. The *OPL_1_* of the air reference cavity is 155 and the *OPL_2_* of the chitosan sensitive cavity is 30.096. Substituting these parameters into Formula Equation (4), the *OPL* ratio is 5.1, and a 4-order HVE FPI with Δ of 4.52 μm is obtained. Using the inner envelope fitting technology mentioned above, the experimental processing of the reflection spectrum was carried out, and the results are shown in Figure 10a.

At this time, because the FSR of the inner envelope is very large, we chose to track the intersection of the inner envelope for spectral demodulation. In order to obtain an RH sensing range that was as large as possible, the intersection of env2 and env5 was selected. The experimental spectrum under each RH value was fitted by inner envelope, and the spectral change is shown in Figure 10b. Figure 10c was obtained by linear fitting according to the wavelength position and RH. In the observed range, the inner envelope response sensitivity was −52.2659 nm/%RH. After linear fitting, the change of the peak position under the same RH did not exceed 0.15 nm. To enhance the reliability and persuasion of the experiment, we also tracked the envelope intersection of env3 and env4 near the edge of the spectrometer, and the spectral changes are shown in Figure 10d. The wavelength and RH were linearly fitted to obtain Figure 10e. At this tracking point, the sensitivity in the process of RH rise was −52.55 nm/%RH and the sensitivity in the process of RH decline was −52.60488 nm/%RH. The linear fitting correlation coefficients obtained at the two tracking points are greater than 0.999, which indicates that the change of inner envelope position has a good linear relationship with humidity.

By tracking the intersections of the inner envelopes, almost identical RH sensitivity was obtained, and its value was recorded as 52.26589 nm/%RH. The RH sensitivity of the single sensitive cavity is 1.58217 nm/%RH, and the absolute value of the sensitivity magnification obtained in the experiment is M = 33.0343. The structural parameters of the sensor were substituted into Equation (13) to calculate the theoretical sensitivity magnification M = 33.292. The experiment is consistent with the theory.

### 3.2. Stability and Repeatability

To measure the stability of the proposed sensor, we carried out spectral monitoring of the sensor at different times under two RH values of 50% RH and 60% RH. In a constant temperature environment of 25 °C, we set the above two RH values and recorded the experimental data every half hour. We observed the position change of the upper envelope peak of the sensor at about 1360 nm within 6 h, and the results are shown in Figure 11. It can be seen that the peak fluctuation of wavelength will not exceed 0.4 nm in the whole tracking time, that is, the fluctuation of RH detection value within 6 h will not exceed 0.25% RH, which is experimentally reasonable and acceptable.

To verify the repeatability, we carried out the above-mentioned RH experimental measurements on the proposed sensor three times and recorded its spectral information. The upper envelope response sensitivity of the spectrum and the response sensitivity of fitting the inner envelope during each experiment are discussed, respectively. The final experimental results are shown in Figure 12. Finally, the response sensitivities of the upper envelope are 1.57906 nm/%RH, 1.57535 nm/%RH, and 1.57350 nm/%RH, respectively. The sensitivities of fitting the inner envelope (the intersection of env2 and env5) are 52.2698 nm/%RH, 52.14919 nm/%RH, and 52.18478 nm/%RH respectively. The relative average deviation of the sensitivity of the upper envelope is 0.1307%, and the relative average deviation of the intersection of the inner envelope is 0.0875%. The results show that the sensor we designed has good experimental repeatability.

### 3.3. Temperature Cross-Sensitivity and Repeatability

Because of the thermal expansion and thermo optic effects of optical fiber materials and chitosan hydrogel materials, there will be some temperature crosstalk between RH sensors based on optical fiber structure.

To explore the influence of temperature on the designed RH sensor, we carried out temperature experimental research on the proposed sensor. We placed the sensor in the above constant temperature and test chamber. At this time, the relative humidity was set to a fixed value of 65% RH, the temperature changed from 25 °C to 100 °C in steps of 15 °C, and each temperature value was maintained for 1 h. We recorded the experimental spectrum during the temperature rise and fall, and the temperature response is shown in Figure 13a. By analyzing the spectral change of the upper envelope, Figure 13b shows that the temperature change will significantly blue shift the wavelength position of the upper envelope. Figure 13c shows the linear fitting curve obtained by tracking the trough position of the upper envelope near 1430 nm. It can be seen that the sensor has a good linear response during the experiment, regardless of the rise or fall of temperature. It is noted that the temperature response sensitivity at this time was −0.045 nm/°C.

The temperature response spectrum is fitted with the inner envelope under the 4-order HVE. When the temperature is 25 °C, the fitted spectrum is consistent with Figure 10a. Here, we chose to track the intersection of the inner envelope env1 and env5 to obtain the spectral change diagram as shown in Figure 13d. Although the upper envelope changes regularly when the temperature changes, the inner envelope deformation is somewhat serious. The reason is that the temperature changes the physical characteristics of the film sensitive cavity and the refractive index of the gas in the air cavity. The changes of these parameters distort the inner envelope to a certain extent. Although its shape changes with the environment, we tracked the position of envelope intersection so that the shape change would not affect our experimental results. Figure 13e was obtained by linear fitting of the intersection position and temperature value. It can be seen that the sensitivity in the process of temperature was −1.48718 nm/°C.

As mentioned previously, under the structural parameters of the main sensor, the sensitive magnification based on the fourth-order harmonic Vernier effect is M = 33.292. The actual magnification measured in the experiment is M = 32.8889 (taking the temperature sensitivity of the intersection of the inner envelope as −1.480 nm/°C). The experimental value is consistent with the theoretical analysis, which shows that the temperature response of the main sensor is also consistent with the theoretical analysis of HVE.

Through the above experimental analysis of temperature response, we found that the designed cascade FPI humidity sensor is not sensitive to temperature. Whether it is upper envelope response sensitivity or inner envelope fitting, the response sensitivity of RH is about 35 times higher than that of temperature, and the cross sensitivity caused by temperature is only 0.028 %RH/°C. Therefore, in applications requiring low temperature response, the accuracy error caused by this part can be ignored. If the temperature crosstalk cannot be ignored in practical applications, the high linearity of our temperature and RH experimental results also shows that it can be compensated by measuring the temperature response sensitivity of the sensor, proving that the sensor based on HVE can overcome the influence of temperature.

### 3.4. Supplementary Experiments

To further verify that the HVE FPI has a great tolerance of Δ and can realize ultra-high sensitivity, we conducted a supplementary experiment. Another HVE FPI senor with a higher *i*-th order and bigger Δ was accomplished by increasing L1 to 479 μm during fabrication, as shown in Figure 14a. L2 was maintained at approximately 20 μm; the actual value was 18.2 μm. According to the previous theoretical analysis, the sensor can introduce the 16-th order HVE (Δ = 8.71 μm). The envelope curve obtained by the inner envelope fitting technology is shown in Figure 13b.

We chose to track the envelope intersection of env1 and env11, and the spectrum of the upper envelope and intersection with RH are shown in Figure 14c,e. The RH sensitivity of the upper envelope and internal envelope are 1.59 nm/%RH and −83.77 nm/%RH, respectively, as shown in Figure 14d,f. The experimental M = −53.98, which is very close to the theoretical value of −52.64. The experimental results prove that, even with an unideal Δ (as large as 8.71 μm), an ultra-high sensitivity of −83.77 nm/%RH, which is 402 times that in [4], and a large *M*-factor (−53.982) can still be achieved by increasing the order through adjusting L1. As the theoretical analysis indicates, an ultra-high sensitivity is achieved, and the requirement of adjustment accuracy of L1 is more relaxed than that of Δ. This is because the adjustment of the interval of the *i*-th order of the entire OPL of FP2 is about 30μm, which is far larger than Δ. Therefore, an HVE FPI is much easier to use to achieve ultra-high sensitivity compared to a TVE FPI.

Table 1 shows a comparison of the performances of the proposed sensor with the previously reported studies for humidity fiber-optic sensors. Due to HEV, the proposed sensor could obtain higher sensitivity compared with the previous work. Our detection range is limited by the insufficient spectrum of the broadband light source.

## 4. Conclusions

In summary, a cascaded FPI fiber sensor based on the HVE has been demonstrated for ultra-high-sensitivity RH detection. The sensitivity *M*-factor in this method depends on the *OPL* mismatch and on the harmonic order of the HVE, effectively solving the problem of the strict *OPL* matching condition of the TVE for achieving a high *M*-factor. The intersection of the internal envelope tracking method allows spectra demodulation to no longer be limited by the size of the FSR of the TVE envelope. A high sensitivity of −83.77 nm/%RH has been experimentally achieved. This work can be useful for subsequent research of ultra-high-sensitivity sensors.

## Figures and Tables

**Figure 1 sensors-22-04816-f001:**
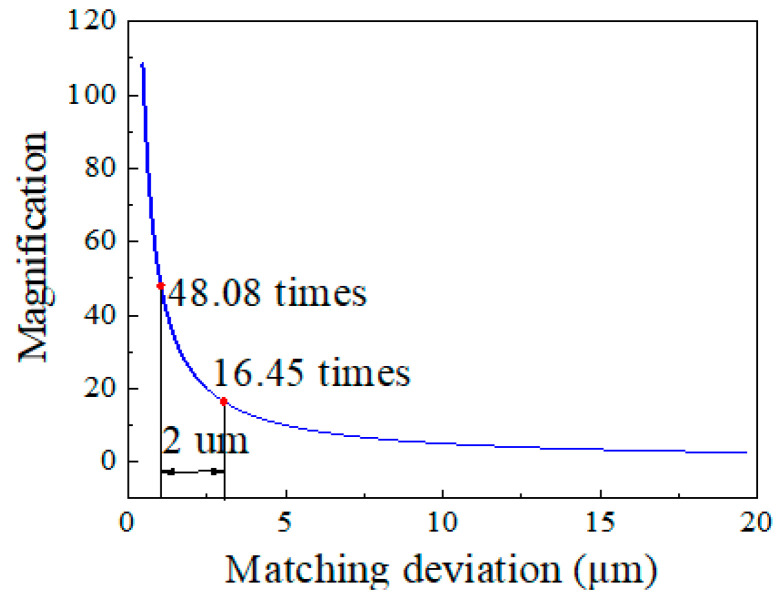
Relationship between *M*-factor and *OPL* matching deviation.

**Figure 2 sensors-22-04816-f002:**
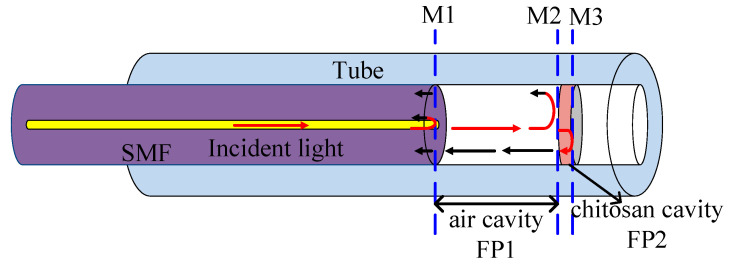
Schematic diagram of the proposed HVE FPI sensor.

**Figure 3 sensors-22-04816-f003:**
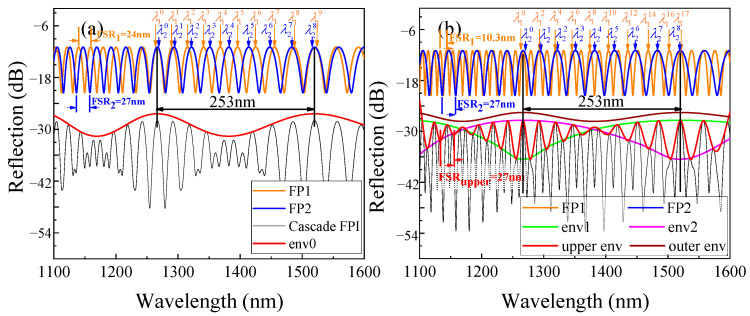
The spectra of FP1, FP2, and cascaded HVE FPI at (**a**) *i* = 0 and (**b**) *i* = 1.

**Figure 4 sensors-22-04816-f004:**
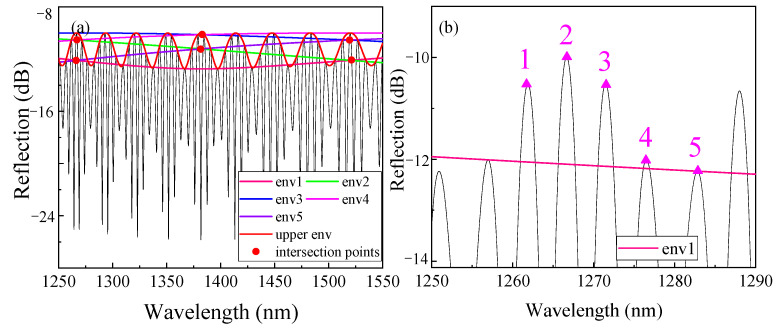
Simulated spectra of the HEV. (**a**) The spectra of FP1, FP2, and cascaded HVE FPI at *i* = 4 and (**b**) locally amplified spectra with spaced points.

**Figure 5 sensors-22-04816-f005:**
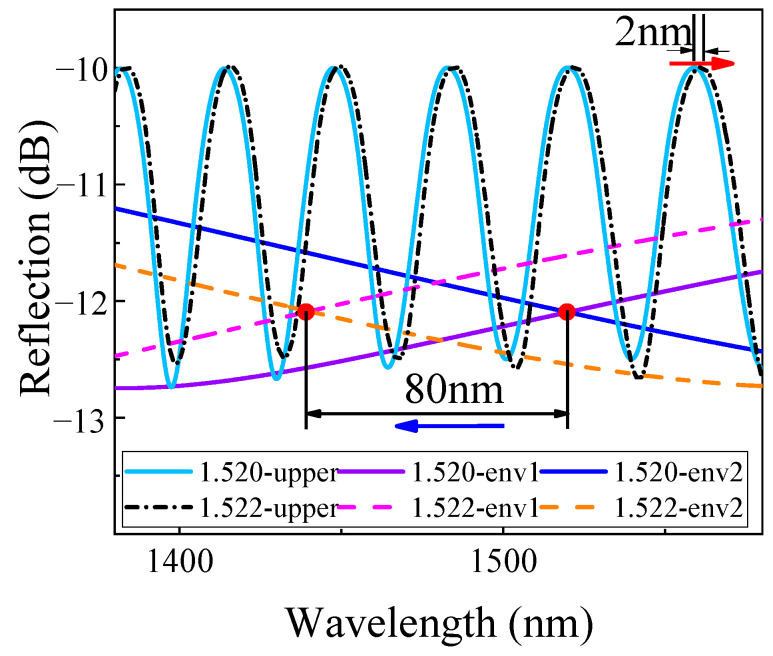
The spectral drifts when n2 changes from 1.520 to 1.522, and *i* = 4.

**Figure 6 sensors-22-04816-f006:**
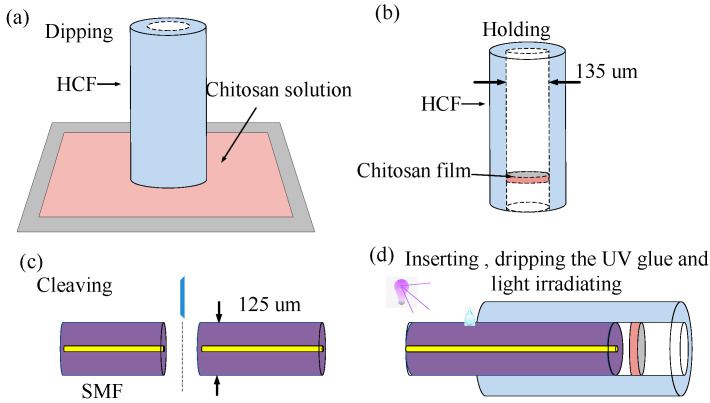
Sketches illustrating the sensor fabrication process—(**a**) dipping the HCF into the chitosan solution; (**b**) holding the HCF to form a thin chitosan film; (**c**) cleaving the SMF; and (**d**) inserting the SMF into the HCF, dripping the UV glue and curing under violet light irradiation.

**Figure 7 sensors-22-04816-f007:**
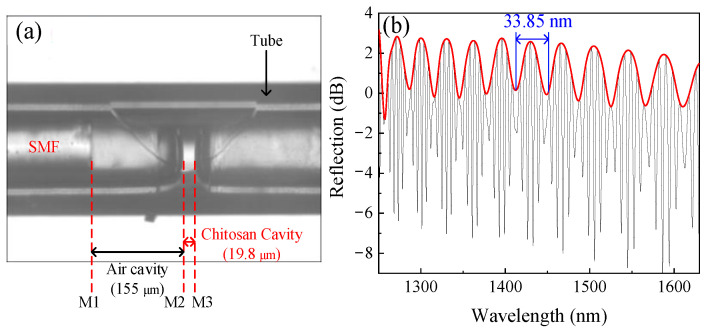
Microscopic image of the sensor and corresponding spectrum. (**a**) Microscopic image of the sensor; (**b**) spectrum of the sensor.

**Figure 8 sensors-22-04816-f008:**
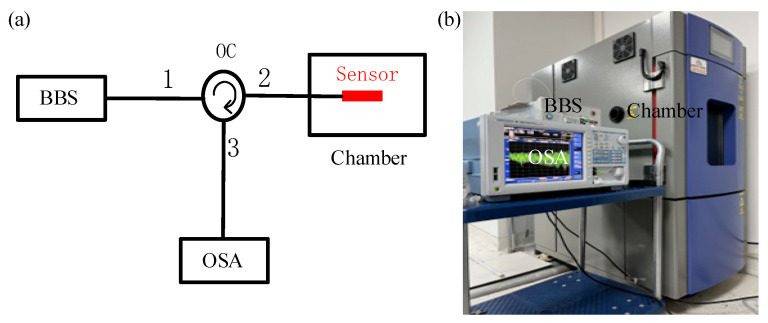
Experimental setup for the RH measurement. (**a**) Structural schematic diagram and (**b**) actual experimental installation.

**Figure 9 sensors-22-04816-f009:**
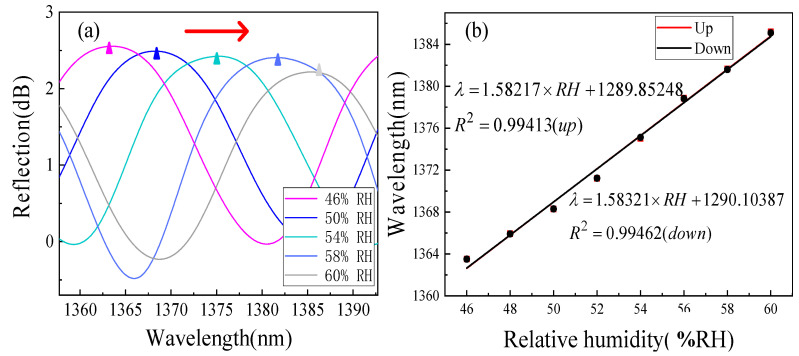
RH response results of the upper envelope. (**a**) Reflection spectra of the sensor at 46–60%RH and (**b**) measured wavelength shift vs. RH rising and falling.

**Figure 10 sensors-22-04816-f010:**
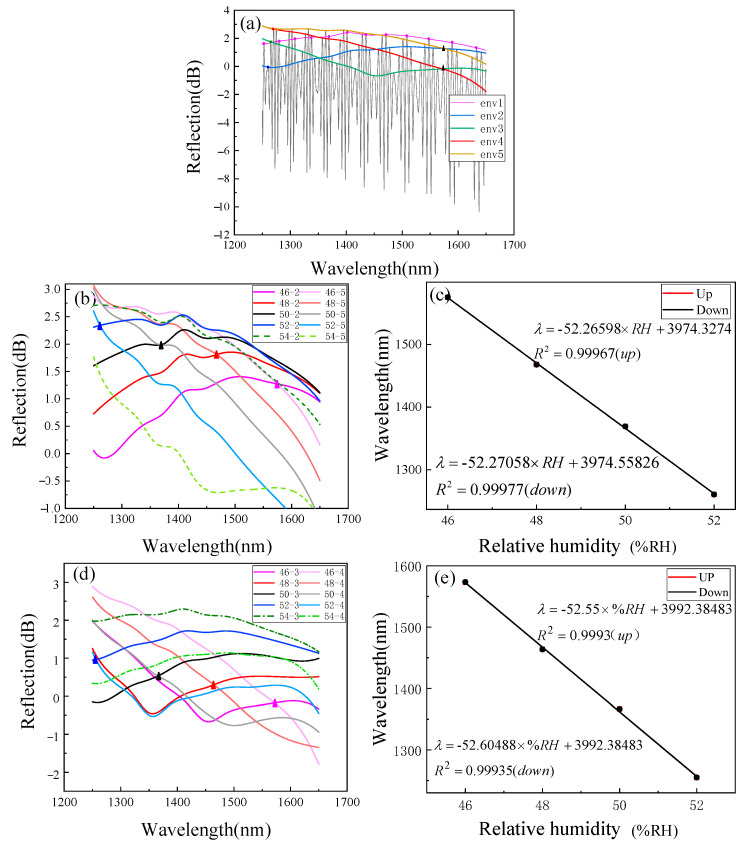
RH response results of inner envelope. (**a**) Spectra of the inner envelope fitting; (**b**) spectral changes of the intersection of env2 and env5; (**c**) the intersection of env2 and env5 has a linear relationship with RH; (**d**) spectral changes of the intersection of Env 3 and env 4; and (**e**) the intersection of env3 and env4 has a linear relationship with humidity.

**Figure 11 sensors-22-04816-f011:**
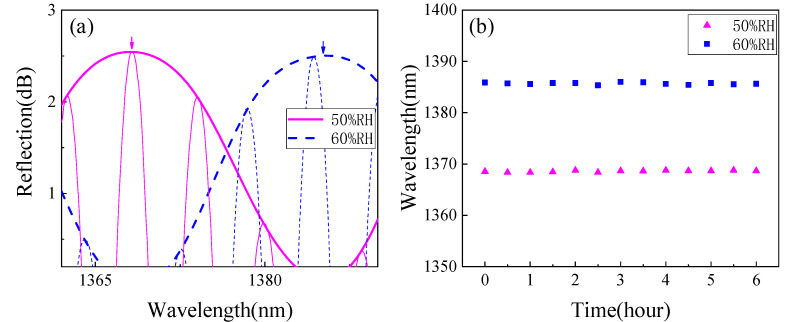
Stability experiment results. (**a**) Spectra at 50% RH and 60% RH and (**b**) stability test results.

**Figure 12 sensors-22-04816-f012:**
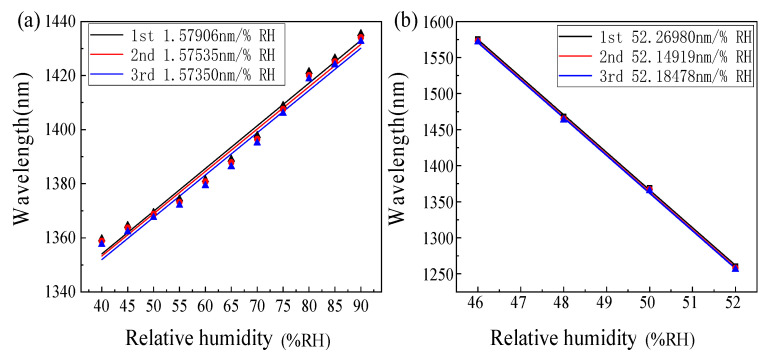
Repeated experiment results. (**a**) Upper envelope response in repeated experiment and (**b**) the inner envelope response was fitted in the repeated experiment.

**Figure 13 sensors-22-04816-f013:**
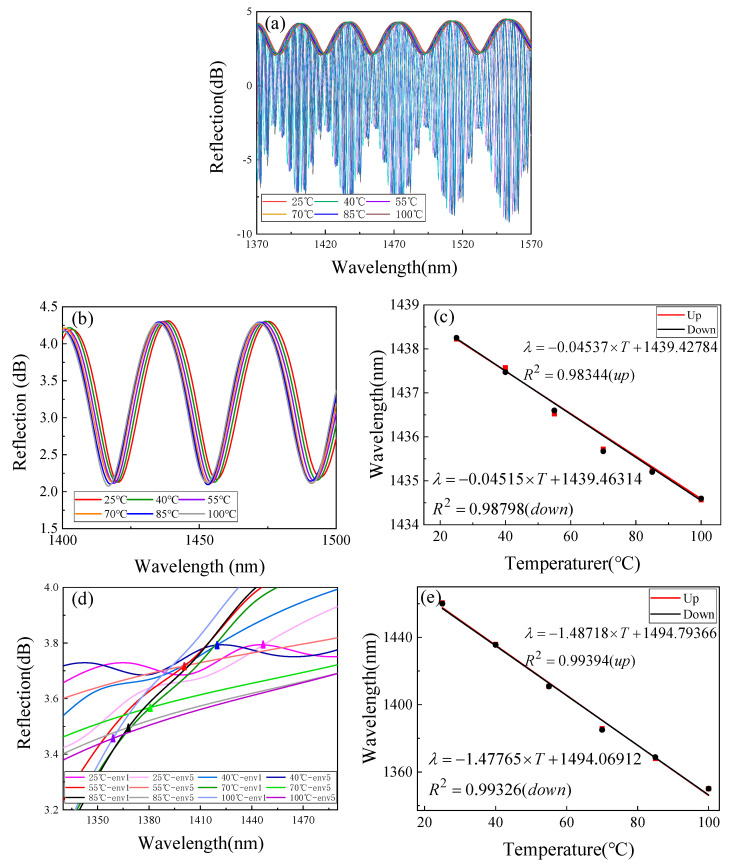
Temperature crosstalk experiment results. (**a**) Temperature response of the sensor; (**b**) spectral variation of the upper envelope; (**c**) the linear relationship between the position of the upper envelope and temperature; (**d**) spectral changes at the intersection of env1 and env5; and (**e**) linear relationship between intersection position and temperature.

**Figure 14 sensors-22-04816-f014:**
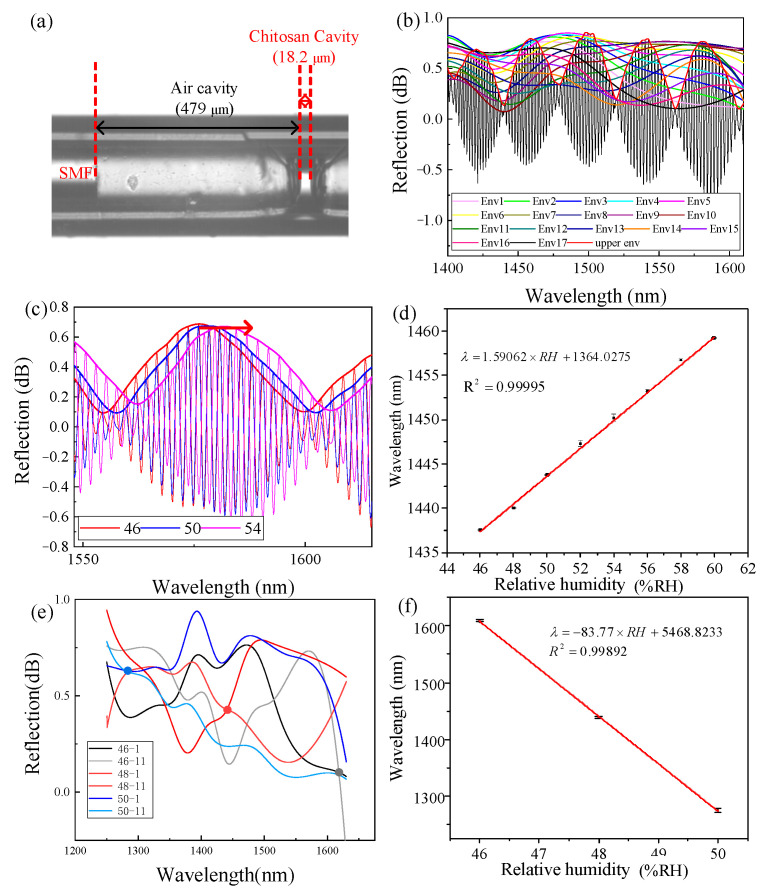
Supplementary experiment results. (**a**) Microscopic image of the sensor; (**b**) inner envelope fitting spectrum; (**c**) spectra of upper envelope change; (**d**) measured wavelength shift vs. RH change of upper envelope; (**e**) spectra of inner envelope change; and (**f**) measured wavelength shift vs. RH change of inner envelope.

**Table 1 sensors-22-04816-t001:** Performance comparison of the proposed sensor and other optical fiber-based humidity sensors reported in literature.

Type	Humidity Sensitivity Material	Range %RH	Sensitivity
FP^4^	Chitosan	20–95	0.13 nm/%RH
FP^12^	Chitosan	35–95	0.28 nm/%RH
FP^11^	Chitosan	30–95	0.081 nm/%RH
FP^15^	Spider dragline silk	58–95	Average 17.2 nm/%RH
FP^12^	Al_2_O_3_	20–90	0.31 nm/%RH
M-Z Interferometric^9^	No coating	50–90	0.02 nm/%RH
M-Z Interferometric^10^	No coating	35–95	−0.083 dB/%RH
Modal Interferometric^15^	SnO_2_	20–90	0.31 nm/%RH
FBG^17^	Pore-Foaming Agent Doped Polyimides	20–90	1.71 pm/%RH
Our work FP	Chitosan	46–50	83.77 nm/%RH

## Data Availability

Not applicable.

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
