# Peer review of "Ultra-High-Sensitivity Humidity Fiber Sensor Based on Harmonic Vernier Effect in Cascaded FPI"

_sensors, 2022, doi:10.3390/s22134816_

Round 1

Reviewer 1 Report

The authors presented a humidity fiber sensor based on the harmonic Vernier effect. While the results and discussion are quite well presented, I have several comments as follows: 

1. The introduction can be more comprehensive as it currently lacks the discussion of prior harmonic Vernier effect works. I would suggest the authors review and discuss more such works, especially [R1], and discuss the novelty of the proposed work. 

2. Part of the contents in section 2, especially the equations look to be referenced from [R1]. I suggest citing [R1] in the relevant texts/equations. 

3. Please discuss the limitation of the harmonic orders. 

Minor comments:

Line 190: Fig. 5 --> Fig. 6?

Line 193: Please change the figure caption to reflect the figures. 

Line 324: senor -> sensor?

R1. Gomes, A.D.; Ferreira, M.S.; Bierlich, J.; Kobelke, J.; Rothhardt, M.; Bartelt, H.; Frazão, O. Optical Harmonic Vernier Effect: A New Tool for High Performance Interferometric Fiber Sensors. Sensors 2019, 19, 5431

Reviewer 2 Report

The article reported by Cheng et al. entitled “ultra-high sensitivity humidity fiber sensor based on harmonic vernier effect in cascaded FPI” present the experimental validation of humidity sensor based on FPI utilizing chitosan as Fabray cavity and exhibits a high sensitivity of −83.73 nm/%RH. All in all, this is a good piece of work supported by experimental validation. The article is a good match to the goal of Sensors, however, there are some concerns that need to be addressed:

1. The state of the art needs to be improved, the introduction section lack literature. There are several papers published in recent years, hence authors are advised to cite the latest articles in the literature survey. Author may find the following article useful: 10.1143/JJAP.21.1509, 10.1109/JPHOT.2021.3069396, 10.1016/j.snb.2021.130154, etc.

2. Authors are advised to cite the original articles used for equations.

3. What is the rule of optimization of thickness of chitosan and cavity length, describe briefly.

4. How the author has chosen the peak to study the shifting, as the spectra give several peak shifts.

5. Fig. 7, can the author show the actual experimental chamber? Also, a detailed discussion is required over the experimental setup in terms of the range of wavelength, type of source, resolution, chamber size, etc. technical detail is required for readers' point of view o reproduce the experiment.

6. The authors should describe the technique to fabricate the sensor, is there any pre functionalization required to deposit the material, and how they fix the tube at a specified position from the fiber to maintain a similar cavity length.

7. It is clear that the chitosan solution was deposited by with dipping and curing method, although it is not clear to me how the author controlled and maintain the cavity length the same every time? How do they fix the thickness of the film? Technical detail is a must. Is it the dip-coating technique author discussed here? if so please discuss the speed pf pulling, dipping, and waiting time.

8. How long is it required to cure the material, what kind of UV and wavelength was used in curing? 

9. How many times the experiment was performed and what was the maximum deviation achieved from each trial. It is advised to show the error bar.

10. The abstract and conclusion show different sensitivity, authors are advised to correct the typo.

11. Presentation and technical merit of the work seem quite satisfactory for the reader but grammar and syntax need to be improved.

Reviewer 3 Report

Authors presented an ultra-high-sensitivity fiber humidity sensor with a chitosan film cascaded Fabry–Perot interferometer (FPI) based on the harmonic Vernier effect. Thus, I believe that revision is necessary to maintain the publication's high quality. 

1.     Author’s should compare the sensing performance of proposed sensor with existing similar sensors in a tabular form to showcase the novelty of proposed work. 

2.     What is the novelty in sensor structure? Similar kinds of sensor structure is reported earlier. 

3.     What is the reason for keeping the thickness of chitosan as 20 μm?

4.     Authors should improve the figure of experimental setup shown in Fig. 7. They should also add the picture of real setup in inset. 

5.     Also, mention about the specifications of experimental instruments used during measurement. 

6.     Author’s should also do the pH testing. 

7.     In the linear plot shown in Fig. 13, there should be error bar to valiadate the experimental results.

Round 2

Reviewer 1 Report

The authors have provided satisfactory revisions to the reviewer's suggestions. I have no further comments. 

Reviewer 2 Report

The author has addressed all the raised concerns positively, hence I recommend its publication in Sensors. Good luck